# GenomeScope 2.0 and Smudgeplot for reference-free profiling of polyploid genomes

T. Rhyker Ranallo-Benavidez [1✉], Kamil S. Jaron [2,3] & Michael C. Schatz[1,4]

An important assessment prior to genome assembly and related analyses is genome profiling, where the k-mer frequencies within raw sequencing reads are analyzed to estimate major genome characteristics such as size, heterozygosity, and repetitiveness. Here we introduce GenomeScope 2.0 (https://github.com/tbenavi1/genomescope2.0), which applies combinatorial theory to establish a detailed mathematical model of how k-mer frequencies are distributed in heterozygous and polyploid genomes. We describe and evaluate a practical implementation of the polyploid-aware mixture model that quickly and accurately infers genome properties across thousands of simulated and several real datasets spanning a broad range of complexity. We also present a method called Smudgeplot (https://github.com/KamilSJaron/smudgeplot) to visualize and estimate the ploidy and genome structure of a genome by analyzing heterozygous k-mer pairs. We successfully apply the approach to systems of known variable ploidy levels in the *Meloidogyne* genus and the extreme case of octoploid *Fragaria × ananassa*.

[1] Johns Hopkins University, Baltimore, MD, USA. [2] University of Lausanne, Lausanne, CH, Switzerland. [3] Swiss Institute of Bioinformatics, Lausanne, CH, Switzerland. [4] Cold Spring Harbor Laboratory, Cold Spring Harbor, New York, NY, USA. ✉email: tbenavi1@jhu.edu

Genome sequencing has become an integral part of modern molecular biology. The majority of the available analysis methods, however, are designed for established model organisms with chromosome-level reference genomes and detailed annotation readily available. In contrast, genome assemblies of non-model organisms are often fragmented, incomplete, or non-existent. Furthermore, model organisms usually have relatively modest complexity, and are typically haploid or diploid species with relatively low genetic diversity and low repetitive content. Conversely, non-model species often have higher ploidy or higher rates of heterozygosity, and thus are substantially more difficult to analyze. As a result, polyploid species or species with other unusual genome structures are greatly underrepresented among genomics studies.

This underrepresentation reduces the generality of biological insights that can be gleaned from such studies. Notably, polyploids are known to be common, especially among plants and fungi. More than 70% of flowering plants are polyploid[1] including many common crops essential for human consumption and use, such as apples, bananas, potatoes, strawberries, and wheat[2]. Higher ploidy levels have also been documented in many fungal species[3]. Polyploidy in animals is less common than in these other taxa, but is far from rare, including many species of frogs[4], fish[5], crustaceans, and molluscs[6], as well as many species of nematodes[7]. The nematode species that are major pests of polyploid crops also happen to be polyploid[8]. More generally, polyploidization events have important consequences to genome evolution[9,10]. Developing tools to analyze fragmented and polyploid genomes is therefore essential for our understanding of how polyploidy affects genome and species evolution[11].

The methods to analyze polyploid genomes typically rely on mapping reads to a haploid reference. However, obtaining a complete haploid reference is usually a challenging task[12] as the assembly often results in mixed ploidy levels among the assembled sequences[13]. Genome assembly has an extra layer of complexity when the basic genomic features of the species are unknown (e.g., size, heterozygosity, and even ploidy). In the context of diploid organisms, several computational approaches have been developed to estimate genome characteristics directly from unassembled sequencing reads, including genome size and heterozygosity[14–16] or repetitiveness and heterozygosity[17]. However, none of these approaches model polyploid genomes.

We previously introduced GenomeScope[18], for reference-free analysis of diploid genomes using a statistical analysis of k-mers in unassembled reads, also called the k-mer spectrum. Here, we present GenomeScope 2.0, which extends this approach with a polyploid-aware mixture model to computationally infer genome characteristics from unassembled sequencing data. GenomeScope 2.0 fits a mixture of negative binomial distributions to the k-mer spectrum of the sequencing data, with additional components to capture k-mers across higher ploidy levels. To further assist in the analysis of species we also develop Smudgeplot, a visualization technique of genome structure to estimate the ploidy, which is often unknown in non-model organisms. We show that these tools quickly and accurately analyze sequencing data from an ensemble of simulated polyploid genomes and from several real polyploid genomes (see Supplementary Table 1 for a summary of the 11 species analyzed). These tools can be used to improve the assessment and interpretation of genome assemblies and will substantially aid future studies of polyploid or otherwise complex genomes.

## Results

**Methods overview**. We have extended the GenomeScope modeling for polyploid genomes. Similar to GenomeScope 1.0[18], GenomeScope 2.0 takes as input the k-mer spectrum, performs a nonlinear least-squares optimization to fit a mixture of negative binomial distributions, and outputs estimates for genome size, repetitiveness, and heterozygosity rates. For example, Fig. 1 shows the k-mer profiles, fitted models, and estimated parameters for diploid *Arabidopsis thaliana* and triploid nematode *Meloidogyne enterolobii*. The diploid has two major peaks at ~22 and 44, and the triploid has three major peaks centered at ~150, 300, and 450. Occasionally, it is difficult to determine whether a peak in the k-mer spectrum is a major peak. For this reason, GenomeScope 2.0 analyzes a transformed k-mer spectrum (see "GenomeScope

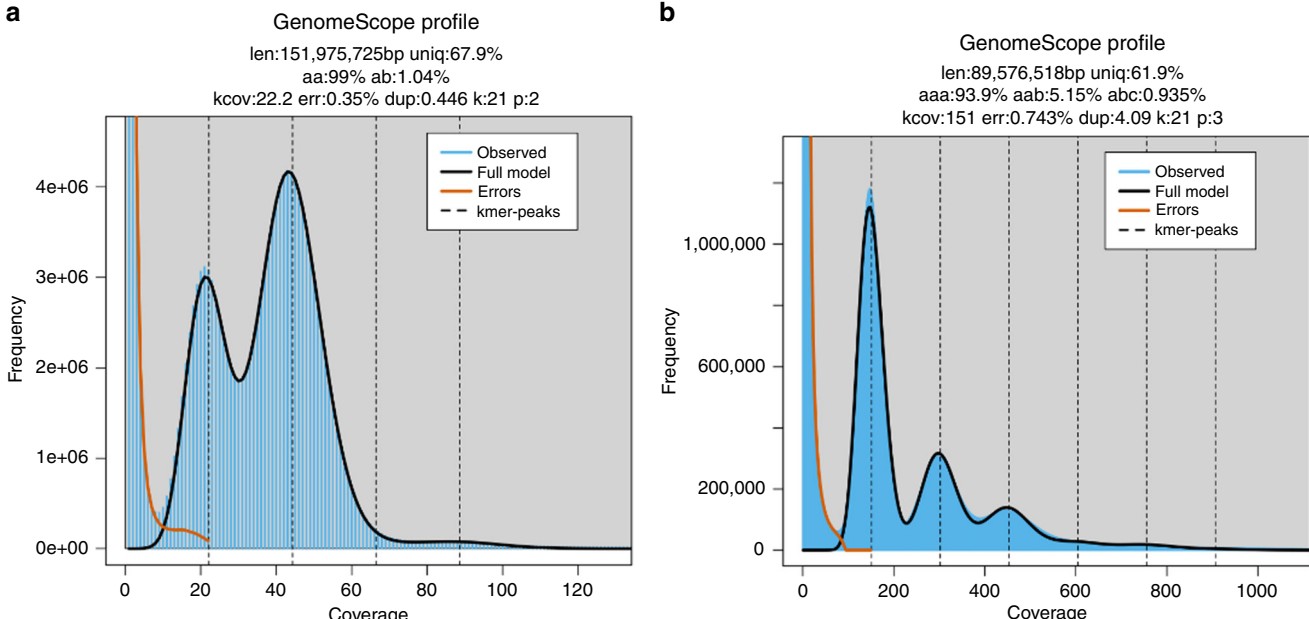

**Fig. 1 GenomeScope plots for heterozygous species.** K-mer spectra and fitted models for (**a**) diploid *Arabidopsis thaliana* and (**b**) triploid *Meloidogyne enterolobii*. Note that the diploid plot has two major peaks, while the triploid plot has three major peaks. Both also have high frequency putative error k-mers with coverage near 1.

Implementation") in which the heights of higher-order peaks are increased. If the ploidy is still uncertain the user may run our Smudgeplot tool (see "Smudgeplot").

Furthermore, the relative heights of the peaks in a k-mer spectrum are proportional to the heterozygosity of the species. For example, for a diploid species, increasing heterozygosity will result in a higher first peak and a lower second peak. For a polyploid species, the relationship is more complicated, but in general increasing heterozygosity will result in a higher first peak and lower subsequent peaks. Lastly, higher coverage peaks of the k-mer spectrum represent increasingly higher copy repetitive sequences in the genomes.

**Simulated polyploid sequencing data**. We first applied GenomeScope 2.0 on 13,704 simulated datasets with varying ploidy (3, 4, 5, and 6), repetitiveness (0, 10, and 20%), and nucleotide heterozygosity rates (0, 0.5, 1, 1.5, and 2% for ploidies 3 and 4; 0, 1, and 2% for ploidies 5 and 6). For each ploidy, we also simulated all the possible topological relationships between the homologous chromosomes. For example, for tetraploid organisms there are two possible topologies (see Fig. 2 for the corresponding representations in Newick notation[19]). For pentaploid organisms there are five possible topologies, and for hexaploid organisms there are sixteen possible topologies (see Supplementary Methods for further explanation).

Each triploid topology consists of two nucleotide heterozygosity forms (e.g., *aab* and *abc*), while each tetraploid, pentaploid, and hexaploid topology consists of three, four, and five heterozygosity forms, respectively. Thus, we simulated 75 triploid datasets (3 repetitiveness values, 5 heterozygosity values for each of the two heterozygosity forms, one topology), 750 tetraploid datasets (3 repetitiveness values, 5 heterozygosity values for each of the three heterozygosity forms, two topologies), 1215 pentaploid datasets (3 repetitiveness values, 3 heterozygosity values for each of the four heterozygosity forms, five topologies), and 11,664 hexaploid datasets (3 repetitiveness values, 3

heterozygosity values for each of the 5 heterozygosity forms, 16 topologies).

For the simulated data, we simulated 15x coverage per homolog and 1% sequencing error, to test GenomeScope 2.0 in relatively poor data quality conditions. Each simulated dataset was created with a generative model using a random 1 Mbp monoploid genome as aprogenitor. To test GenomeScope's robustness on genomes of varying size, we also simulated using progenitor genomes of size 1, 10, 100 Mbp, and 1 Gbp. The mean absolute errors of the estimated parameters on the simulated datasets are shown in Table 1, which demonstrate that GenomeScope 2.0 is highly accurate. For the full results, see Supplementary Data 1.

We then performed more specific testing to validate GenomeScope 2.0's performance at predicting nucleotide divergence, repetitiveness, and length. Specifically, for each of these three parameters, we held the others constant, and varied only the parameter of interest:

(1) For nucleotide divergence, we systematically evaluated across 0–25% in 0.5% increments, for a total of 51 values. We used a 100 Mbp progenitor genome, 15x coverage per homolog, and 1.0% sequencing error. Figure 3 shows the difference between the estimated and true nucleotide divergence as a function of the true nucleotide divergence, for ploidies 2, 3, 4, 5, and 6.

(2) For repetitiveness, we evaluated a parameter sweep from 0-50% in 1% increments, for a total of 51 values. We used a 100 Mbp progenitor genome, 15x coverage per homolog, and 1.0% sequencing error. Figure 4 shows the difference between the estimated and true repetitiveness as a function of the true repetitiveness, for ploidies 1, 2, 3, 4, 5, and 6.

(3) For genome length, we evaluated progenitor genomes of size 1, 10, 100 Mbp, and 1 Gbp. We sequenced 15x coverage per homolog, and 1.0% sequencing error. Figure 5 shows the relative error in the length $\left( \frac{\text{Length}_{\text{Estimated}} - \text{Length}_{\text{True}}}{\text{Length}_{\text{True}}} \right)$ as a function of the true length (log scale), for ploidies 1, 2, 3, 4, 5, and 6.

When compared with GenomeScope 1.0, GenomeScope 2.0 is more robust and accurate, especially on low coverage diploid data. Specifically, GenomeScope 1.0 failed to converge for 35 of the 51 simulated heterozygosity datasets, converged to the wrong peak due to low sequencing coverage for 15 of the datasets, and produced accurate results for only one dataset. GenomeScope 1.0 failed to converge on 2 of the 51 simulated repetitiveness datasets and converged to the wrong peak for the other 49 datasets. Lastly, GenomeScope 1.0 failed to converge for three of the four simulated length datasets and produced inaccurate results for the other dataset. Based on these results, we encourage all users to use GenomeScope 2.0 for diploid datasets.

Finally, we validated Smudgeplot on simulated data. In each case, we simulated 25x coverage per homolog and 1% sequencing error using a random 10 Mbp monoploid genome as a "progenitor". We simulated both the allotetraploid and autotetraploid

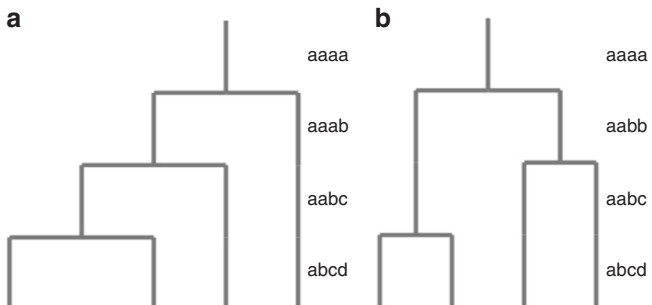

**Fig. 2 Autotetraploid and allotetraploid topologies. a** The autotetraploid topology, notated as (, (, (, ))); in Newick notation, corresponds to the following nucleotide heterozygosity forms: aaaa, aaab, aabc, abcd. **b** The allotetraploid topology, notated as ((, ), (, )); in Newick notation, corresponds to the following nucleotide heterozygosity forms: aaaa, aabb, aabc, abcd.

**Table 1 Mean absolute errors of parameters on simulated polyploid datasets.**

| Mean absolute errors | Triploid | Tetraploid | Pentaploid | Hexaploid |
|---|---|---|---|---|
| Repetitiveness (*d*) | $2.29 \times 10^{-3}$ | $6.61 \times 10^{-3}$ | $9.64 \times 10^{-3}$ | $1.67 \times 10^{-2}$ |
| Nucleotide divergence | $3.58 \times 10^{-4}$ | $7.38 \times 10^{-4}$ | $1.13 \times 10^{-3}$ | $3.76 \times 10^{-3}$ |
| Monoploid length | 2182 bp | 4320 bp | 5138 bp | 7969 bp |

Simulated datasets include 75 simulated triploid datasets, 750 simulated tetraploid datasets, 1215 simulated pentaploid datasets, and 11,664 simulated hexaploid datasets. Nucleotide divergence refers the proportion of loci along the genome for which the nucleotides across all the homologs are not all the same.

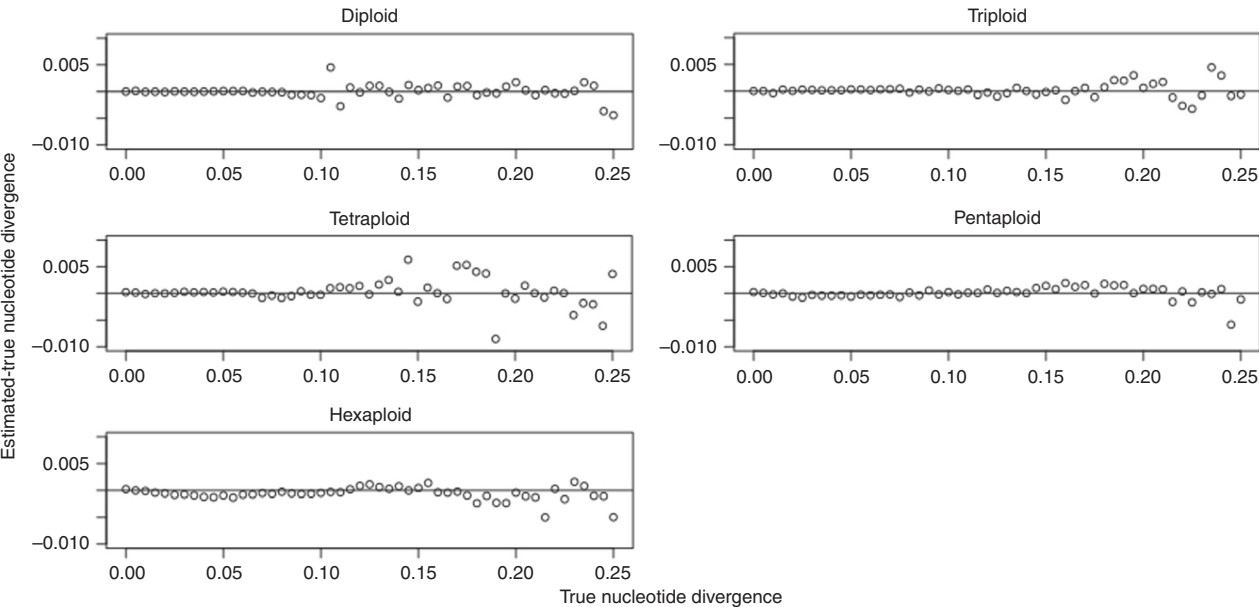

**Fig. 3 Nucleotide divergence parameter sweep.** Results shown for diploid, triploid, tetraploid, pentaploid, and hexaploid simulated datasets.

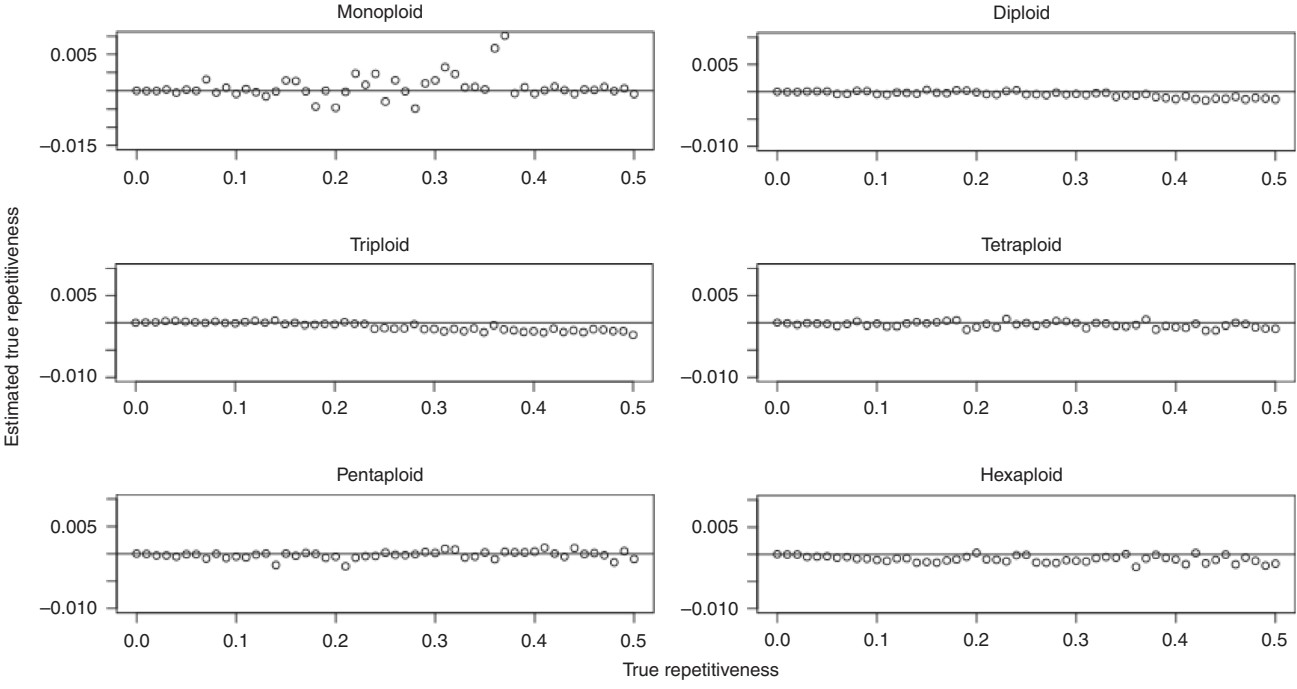

**Fig. 4 Repetitiveness parameter sweep.** Results shown for monoploid, diploid, triploid, tetraploid, pentaploid, and hexaploid simulated datasets.

topologies for ploidy 4, and a single topology for ploidies 2, 3, 5, and 6. For nucleotide divergence, we systematically evaluated across 0.5–25% in 0.5% increments while holding the repetitiveness constant at 10%, for a total of 50 values. For repetitiveness, we evaluated a parameter sweep from 0-50% in 1% increments while holding the nucleotide divergence constant at 2.0%, for a total of 51 values.

For the nucleotide divergence sweep, Smudgeplot correctly estimates ploidy for the diploid simulated data over all heterozygosity values, for the triploid data up to 24.0% heterozygosity, for the allotetraploid data up to 18.0% heterozygosity, for the autotetraploid data up to 23.5% heterozygosity, for the pentaploid data up to 24.0% heterozygosity, and for the

hexaploid data up to 24.0% heterozygosity. Above these heterozygosity thresholds, Smudgeplot underestimates the ploidy due to the k-mers in a k-mer pair being more than one nucleotide different and thus not identified. For the full results, see Supplementary Table 3.

For the repetitiveness sweep, Smudgeplot correctly estimates ploidy for the diploid simulated data up to 39% repetitiveness, for the triploid data up to 38% repetitiveness, for the allotetraploid data up to 43% repetitiveness, for the autotetraploid data up to 38% repetitiveness, for the pentaploid data over all repetitiveness values, and for the hexaploid data over all repetitiveness values. Above these repetitiveness thresholds, Smudgeplot overestimates the ploidy due to the signal from repetitive k-mers dominating

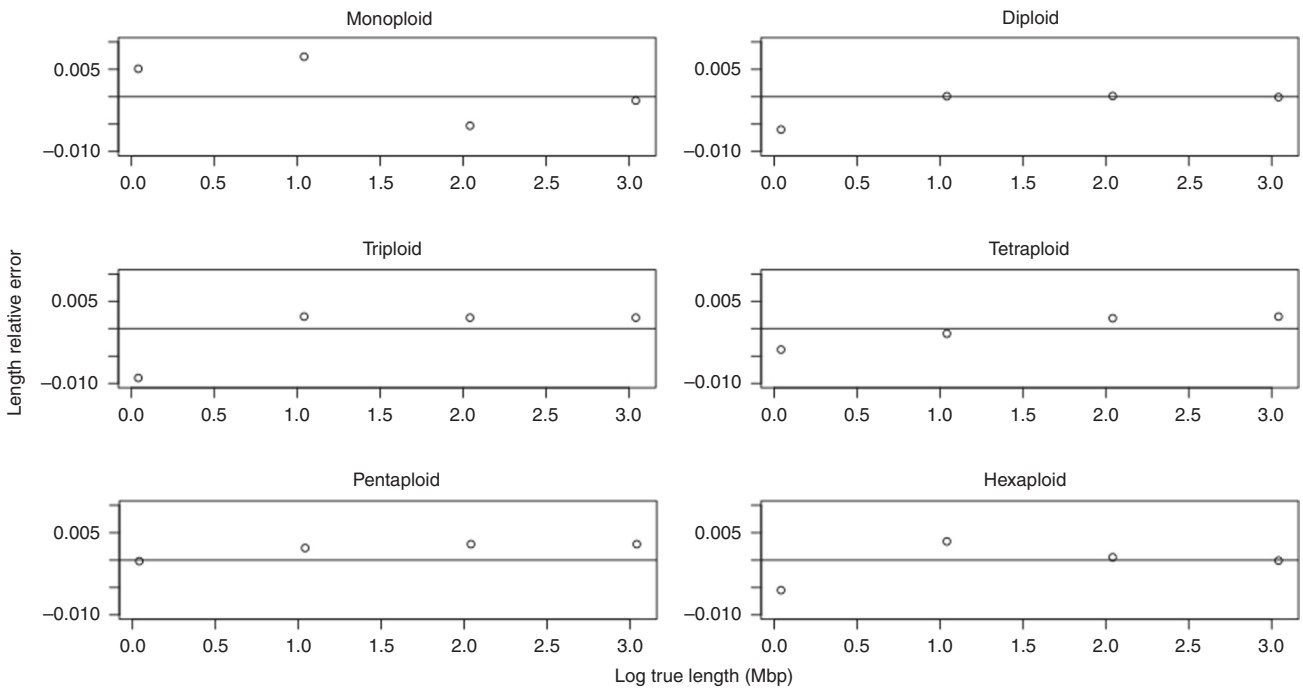

**Fig. 5 Length parameter sweep.** Results shown for monoploid, diploid, triploid, tetraploid, pentaploid, and hexaploid simulated datasets.

**Table 2 Summary of polyploid genomes analyzed.**

| Common name | Species name | Estimated genome size | Assembly size |
|---|---|---|---|
| Coastal redwood | *Sequoia sempervirens* | 27.0 Gbp | 26.5 Gbp |
| Cotton | *Gossypium barbadense* | 2.293 Gbp | 2.267 Gbp[27] |
| Cotton | *Gossypium hirsutum* | 2.349 Gbp | 2.347 Gbp[27] |
| Marbled crayfish | *Procambarus virginalis* | 9.5 Gbp | 3.3 Gbp[21] |
| Root-knot nematode | *Meloidogyne arenaria* | 290.4 Mbp | 163.7 Mbp[7] |
| Root-knot nematode | *Meloidogyne enterolobii* | 268.7 Mbp | 162.4 Mbp[7] |
| Root-knot nematode | *Meloidogyne floridensis* | 201.7 Mbp | 74.9 Mbp[7] |
| Root-knot nematode | *Meloidogyne incognita* | 207.4 Mbp | 122.0 Mbp[7] |
| Root-knot nematode | *Meloidogyne javanica* | 280.2 Mbp | 142.6 Mbp[7] |
| Potato | *Solanum tuberosum* | 3.0 Gbp | 778.7 Mbp[29] |
| Wheat | *Triticum aestivum* | 14.1 Gbp | 15.34 Gbp[23] |

The genome size refers to the polyploid genome size that is estimated by GenomeScope 2.0. The assembly size for the coastal redwood is from the Redwood Genome Project.

the signal from heterozygous k-mers. For the full results, see Supplementary Table 4.

**Real polyploid sequencing data**. We then applied GenomeScope 2.0 on the real polyploid genomes listed in Supplementary Table 1. Table 2 shows a summary of the GenomeScope estimates for polyploid genome size (see Supplementary Table 2 for the full GenomeScope results). Here, we highlight a few notable results from this analysis, and the complete GenomeScope and Smudgeplot plots are available as Supplementary Figs. 1–23.

Coastal redwoods (*Sequoia sempervirens*) are evergreen trees that can reach towering heights and are some of the longest living organisms on Earth. *Sequoia sempervirens* is known to be hexaploid, with recent evidence suggesting that it is an autohexaploid[20]. This aligns with the Smudgeplot analysis, which inferred a triploid ploidy for these data, which come from the haploid megagametophyte extracted from a seed. Furthermore, the genome size of the coastal redwood is larger than the human genome, with a recent assembly by the Redwood Genome Project

spanning 26.5 Gbp. The estimated genome size of the coastal redwood output by GenomeScope is 27.0 Gbp, revealing great concordance with the recent assembly (see Supplementary Figs. 1 and 2).

Marbled crayfish (*Procambarus virginalis*) are freshwater crustaceans that undergo parthenogenetic reproduction, in which a female gamete develops into an individual without fertilization. Based on a Smudgeplot analysis, we inferred the ploidy to be triploid, which aligns with the current understanding of this organism[21]. We ran GenomeScope 2.0 with a triploid model to estimate the genome characteristics. Specifically, GenomeScope estimates a polyploid genome size of 9.7 Gbp, while the current assembly spans 3.3 Gbp (see Supplementary Figs. 7 and 8). It is clear that the assembly only spans one homolog of the triploid genome.

Root-knot nematodes (*Meloidogyne arenaria, Meloidogyne enter-olobii, Meloidogyne floridensis, Meloidogyne incognita,* and *Meloidogyne javanica*) are parasitic roundworms that infect the roots of plants. Based on Smudgeplot analyses, we inferred that *Meloidogyne enterolobii, Meloidogyne floridensis,* and *Meloidogyne incognita*

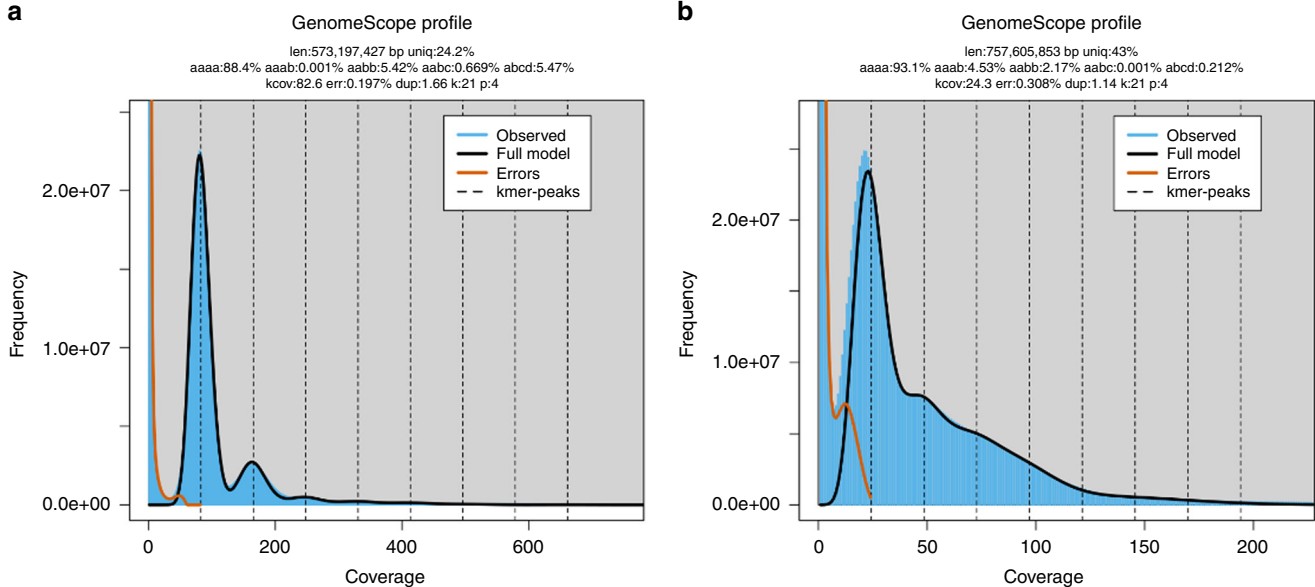

**Fig. 6 Allotetraploid and autotetraploid GenomeScope plots.** K-mer spectra and fitted models for (**a**) allotetraploid *Gossypium barbadense* and (**b**) autotetraploid *Solanum tuberosum*. Note that the allotetraploid plot has *aaab* < *aabb*, while the autotetraploid plot has *aaab* > *aabb*.

were triploid, while *Meloidogyne arenaria* and *Meloidogyne javanica* were tetraploid. Running GenomeScope 2.0 with the corresponding ploidies, we determined estimates for the genome characteristics. For the five root-knot nematodes the GenomeScope estimates for genome size are 1.65–2.69 times larger than the current best assemblies[7], suggesting the assemblies have partially collapsed the homologous chromosomes (see Supplementary Figs. 9–18).

Bread wheat (*Triticum aestivum*) is an allohexaploid which consists of three subgenomes[22]. A Smudgeplot analysis inferred that the ploidy was diploid, because the individual subgenomes are highly divergent from each other. Specifically, if the homologous k-mers from different subgenomes are highly divergent (more than one SNP different), while the homologous k-mers from the same subgenome are only one SNP different, and then we would expect to see three k-mer pairs. Each of these pairs would have an estimated sum of coverages of $2\lambda$ and an estimated relative minor coverage of $\frac{1}{2}$, and would thus be interpreted by Smudgeplot as coming from the genomic structure *AB*. The current best assembly spans 15.34 Gbp[23], while the GenomeScope estimate is 14.1 Gbp (see Supplementary Figs. 21 and 22).

**Allotetraploid vs. autotetraploid.** One important application of GenomeScope is to distinguish between allotetraploid and autotetraploid species based on the distinct patterns of nucleotide heterozygosity rates that occur. For example, it is known in cotton that during meiosis homologous chromosomes from the same subgenome form bivalents and preferentially pair with each other[24]. This phenomenon is also prominent in many other allotetraploid species[25]. Thus, for allotetraploids we would expect a high proportion of *aabb* and a low proportion of *aaab* since preferential pairing would ensure that two homologs from the first subgenome and two homologs from the second subgenome are present after recombination. Conversely, it is known in potato that during meiosis the majority of cells contain quadrivalents[26]. In this case, after recombination an individual might have 0, 1, 2, or 3 homologs from a given subgenome. Thus, *aaab* would be expected to be more prominent than *aabb* since it is more likely that there are one or three copies of a subgenome rather than exactly two copies of a subgenome (see Supplementary Methods for a more in-depth discussion).

For cotton and potato, we see that the GenomeScope estimates for nucleotide heterozygosity rates follow these expectations (see Fig. 6). For the two allotetraploid cotton species, *aaab* is estimated to be ~0 and *aabb* is estimated to be >5%. The estimated genome size is also highly accurate, and GenomeScope estimates the polyploid genome lengths to be 2.293 and 2.349 Gbp, while the current best assemblies span 2.267 and 2.347 Gbp, respectively[27] (see Supplementary Figs. 3–6). For potato[28], *aaab* is greater than *aabb* as we would expect after recombination. Here, the estimated genome size is approximately three times larger than the current best assembly (3.0 Gbp vs. 778.7 Mbp) (see Supplementary Figs. 19 and 20). This is expected since the assembly was filtered to form a pseudo-haploid representation that reports a single homolog[29]. Thus, the GenomeScope estimates can determine whether a novel polyploid organism is an allopolyploid or autopolyploid.

## Discussion

We have shown on simulated and real datasets that Genome-Scope 2.0 is able to quickly and accurately estimate the genomic characteristics of polyploid organisms without a reference genome. The core of GenomeScope 2.0 is a polyploid model using the Möbius inversion formula which accounts for the k-mers occurring at higher ploidy levels. Users provide the k-mer spectrum as input and GenomeScope performs a nonlinear optimization using the Levenberg–Marquardt algorithm. We have also introduced Smudgeplot as a visualization and analysis technique that can be used to reveal the structure of a novel species. The core of this analysis is the identification and statistical analysis of k-mer pairs that differ by exactly one nucleotide.

The coverage of the dataset must be sufficient for these methods to resolve the error peak with the haploid peak. In general, having at least 15x coverage per homolog for Genome-Scope and 25x coverage per homolog for Smudgeplot is required. Currently, GenomeScope and Smudgeplot only support low error short read sequencing. Future work remains to extend these techniques for single molecule sequencing with high error rates that currently prevent k-mer based analysis. Smudgeplot works well under moderate heterozygosity and repetitiveness where the signal from heterozygous k-mer pairs is stronger than the signal from repetitive k-mer pairs. For eight of the real polyploid

organisms analyzed, Smudgeplot produces an accurate estimate of ploidy. Species with extreme heterozygosity and high repetitiveness, such as cotton and wheat, can confuse a Smudgeplot analysis. Another example is the diploid *Fragaria iinumae* strawberry genome, where more k-mer pairs come from the "AABB" smudge than from the "AB" smudge, which leads to the incorrect inference of tetraploidy (see Supplementary Fig. 23). Upon further analysis, Smudgeplot is correctly finding k-mer pairs in the genome, though they actually represent repetitive k-mer pairs, not k-mer pairs at a higher ploidy level. However, GenomeScope results reveal very low levels of heterozygosity and high rates of duplications, which highlight that using these tools in conjunction with one another can help unravel the properties of a genome.

In addition, polyploid species, especially allopolyploids, often have highly divergent genomic copies (e.g., >12% different at the nucleotide level). Thus, one limitation of using a k-mer-based technique is that in these cases too few k-mers may actually be shared between the homologous copies. This can lead Smudgeplot to infer diploidy even for polyploid species. However, in these cases the divergence of the homologs may be so high that they will be separated during the assembly process. The polyploidy will then very likely be revealed by standard genome quality assessment of conserved single-copy orthologs (BUSCO)[30].

It is important to run GenomeScope with the correct value for the ploidy parameter. If $p$ is greater than the true value this can lead to overfitting where the model contains a greater number of negative binomial distributions than is necessary. In real polyploid data, especially for highly heterozygous genomes, it is difficult to know a priori whether shorter higher-order peaks truly represent the data or whether they are due to an incorrect ploidy parameter. In addition, in order to accurately estimate genome size for highly repetitive genomes, it is important to create a k-mer histogram that is not truncated. By default, KMC and Jellyfish truncate the histogram at 10,000. We suggest running these tools without a maximum counter. The model fit that is output by GenomeScope can also be used to identify poor fit or incomplete datasets. However, in general, the best indicator of a good model fit is inspecting the plots to ensure the model matches the empirical data across the distribution.

Even with these caveats, GenomeScope and Smudgeplot are able to rapidly and accurately infer genomic properties for large, highly heterozygous, and polyploid genomes. GenomeScope accurately predicts genomic properties for the nearly 9 Gbp coastal redwood genome, for the highly heterozygous allotetraploid cotton genomes, and for the hexaploid wheat genome. Furthermore, GenomeScope is able to distinguish between allopolyploid and autopolyploid species, which can help researchers gain valuable biological insights for novel organisms without needing to perform costly experiments. Finally, Smudgeplot is able to correctly predict ploidy even in the extreme case of octaploid *Fragaria × ananassa*. These tools will open up future analysis of complex organisms that are underrepresented in current genomics pipelines.

## Methods

**GenomeScope diploid model**. GenomeScope 1.0 statistically analyzes the k-mer profile and fits a mixture of four negative binomials, the first two representing unique heterozygous and homozygous k-mers, and the next two representing two-copy heterozygous and homozygous k-mers. For example, Fig. 1a shows the k-mer profile, fitted model, and estimated parameters for a highly heterozygous diploid *Arabidopsis thaliana* representing an F1 cross between two divergent accessions (Col-0 × Cvi-0)[31].

The four negative binomials are equally spaced apart and occur at $\lambda$, $2\lambda$, $3\lambda$, and $4\lambda$, where $\lambda = 22.2$ is the average k-mer coverage for the diploid genome. More generally, the $i$th peak corresponds to the contributions from k-mers that occur exactly $i$ times in the diploid genome. It should be noted that although

GenomeScope does not fit negative binomials for repetitive regions that occur more than twice, this does not greatly affect the fit on the peaks corresponding to less repetitive regions. This is because the proportion of the genome modeled by a given copy number repeat typically follows a Zeta distribution and hence quickly falls off[32].

The underlying GenomeScope 1.0 model is given by:

$$f(x) = G \sum_{i=1}^{4} \alpha_i NB\left(x, i\lambda, \frac{i\lambda}{\rho}\right), \tag{1}$$

where $f(x)$ is the k-mer spectrum (i.e., the frequency of the k-mers at coverage depth $x$) and $G$ is the number of distinct k-mers (i.e., repetitive k-mers are counted only once) in the *monoploid* genome. Within polyploids, the basic chromosome set from which the other sets are derived is called the monoploid chromosome set, while the chromosomes present in the gametes of a species constitute the haploid chromosome set. Thus, the monoploid genome consists of a single chromosome set, while the haploid genome typically consists of half of the total number of chromosome sets[33]. Under this model, $\alpha_i$ is, for a single distinct k-mer of the monoploid genome, the expected frequency contribution of the corresponding k-mers across the two homologs to peak $i$ of the k-mer spectrum, $NB(x, \mu, size)$ is the negative binomial distribution that approximates the sequencing coverage with mean $\mu$ and dispersion parameter $size$, $\lambda$ is the average k-mer coverage for the diploid genome, and $\rho$ is a bias parameter proportional to PCR duplication and other sequencing biases.

The next crucial step for the model is to mathematically determine the $\alpha_i$ values in terms of the repetitiveness, heterozygosity, and k-mer length. In the diploid case, we have:

$$
\begin{aligned}
\alpha_1 &= (1-d)(2(1-(r_{aa})^k)) + d(2((r_{aa})^k)(1-(r_{aa})^k) + 2(1-(r_{aa})^k)^2) \\
\alpha_2 &= (1-d)((r_{aa})^k) + d((1-(r_{aa})^k)^2) \\
\alpha_3 &= d(2((r_{aa})^k)(1-(r_{aa})^k)) \\
\alpha_4 &= d(((r_{aa})^k)^2),
\end{aligned}
\tag{2}
$$

where $d$ is the proportion of distinct k-mers of the monoploid genome that occur twice, $r_{aa}$ is the homozygosity rate, and $k$ is the k-mer length.

**GenomeScope polyploid model**. To account for the higher ploidy levels in polyploid organisms, the underlying GenomeScope 2.0 model now fits $2 \times p$ negative binomial distributions, where $p$ is the ploidy, according to:

$$f(x) = G \sum_{i=1}^{2p} \alpha_i NB\left(x, i\lambda, \frac{i\lambda}{\rho}\right). \tag{3}$$

Similar to the diploid case, each of the $2p$ negative binomials are equally spaced apart and occur at $\lambda$, $2\lambda$, ..., and $2p\lambda$, where $\lambda$ is the average k-mer coverage of the polyploid genome. Again, the $i$-th peak corresponds to the contributions from k-mers that occur exactly $i$ times in the polyploid genome.

The next step for the model is to mathematically determine the $\alpha_i$ values in terms of the ploidy, repetitiveness, heterozygosity, and k-mer length. In the polyploid case, this calculation is much more involved and requires utilizing the Möbius inversion formula on partially ordered sets, a classical combinatorics theorem[34]. For the derivation of this calculation, please refer to Supplementary Methods.

**GenomeScope implementation**. In order to determine the parameters that best fit the input data, GenomeScope uses a nonlinear least-squares minimization technique. While GenomeScope 1.0 uses the `nls` function in R based on the Gauss–Newton algorithm, GenomeScope 2.0 instead uses the `nlsLM` function. `nlsLM` utilizes the Levenberg–Marquardt algorithm, with support for lower and upper parameter bounds. Like the Gauss–Newton method, the Levenberg–Marquardt algorithm starts from an initial naive estimate and performs an iterative procedure to update the parameters. However, Levenberg–Marquardt introduces a damping parameter that is adjusted as the iterative process continues, making it more robust. Notable, in many simulated and real datasets with higher ploidy, the `nlsLM` function is able to converge, while the `nls` function is not.

For datasets with high heterozygosity and/or high ploidy, the k-mer spectrum does not show clearly defined higher-order peaks. In these cases, fitting to the transformed k-mer spectrum improves the model's ability to converge. We define the transformed k-mer spectrum as a plot of frequency times coverage ($y$-axis) versus coverage ($x$-axis) instead of the typical frequency versus coverage. Transforming the k-mer spectrum effectively increases the heights of higher-order peaks, overcoming the effect of high heterozygosity. This increases the signal in the higher-order peaks, especially the homozygous peak, which allows the model to converge. Even for datasets with low heterozygosity and low ploidy, we find fitting to the transformed k-mer spectrum yields accurate results. Consequently, GenomeScope 2.0 now by default fits to the transformed k-mer spectrum, and the mathematical equation for the model used during the nonlinear optimization is adjusted accordingly. After the fitting process, GenomeScope 2.0 outputs the estimated parameters along with four plots of the best fit model overlaying the k-

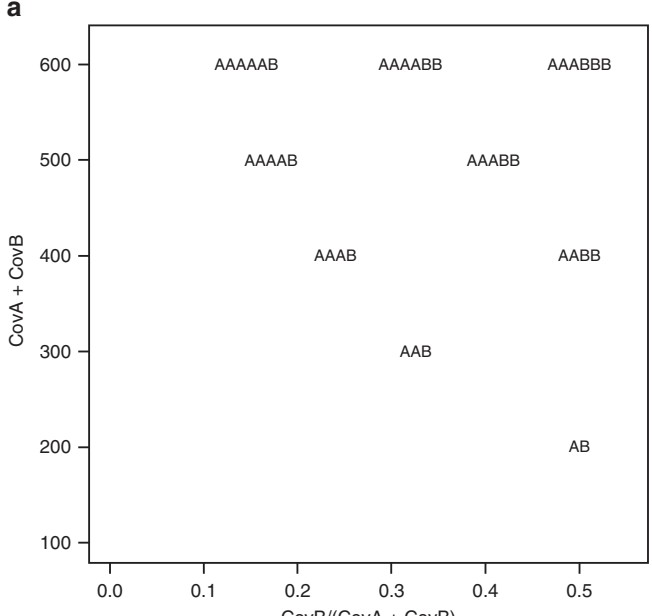

**Fig. 7 Smudgeplot genomic structure locations.** Coordinates of individual genomic structures for a genome with monoploid coverage = 100 in **a** 2D space of coverage sums versus coverage ratios and in **b** a table of coordinates.

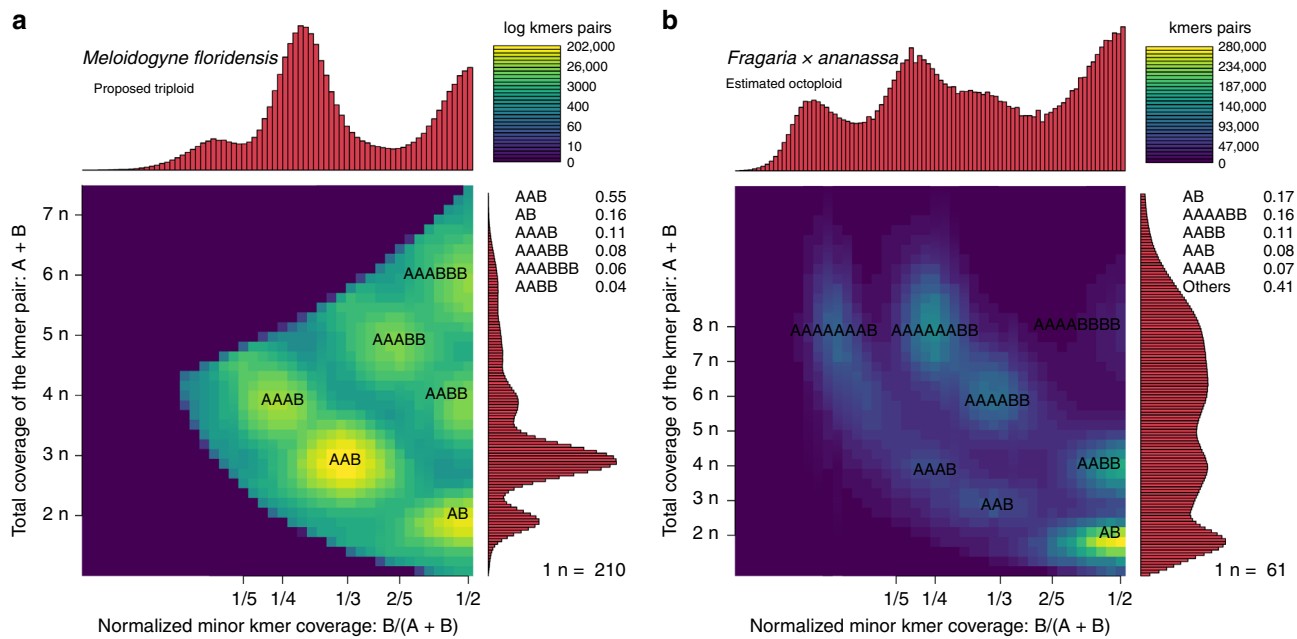

**Fig. 8 Smudgeplots on real datasets.** Smudgeplots for (**a**) the triploid root-knot nematode *Meloidogyne floridensis* and (**b**) the octaploid strawberry *Fragaria × ananassa*.

mer spectrum: (1) untransformed linear, (2) untransformed log, (3) transformed linear, and (4) transformed log.

**Smudgeplot**. GenomeScope 2.0 is able to accurately analyze organisms given a known ploidy. However, in many cases researchers studying a novel organism may not know the ploidy a priori. For this reason, we have implemented Smudgeplot to visualize genome structure and infer ploidy directly from the k-mers present in sequencing reads.

For this method, we take as input the set of sequenced k-mers, such as the k-mer frequency files produced by KMC[35] or jellyfish[36]. Then, we search for all pairs of k-mers that differ at exactly one nucleotide through a systematic scan of all input k-mers. To avoid pairing k-mers produced by sequencing errors with

genomic k-mers, we search only those k-mers which exceed a coverage threshold and assume that such k-mers represent real genomic k-mers. Given how many possible k-mers exist for sufficiently large $k$ (e.g., over four trillion for $k = 21$), it is very unlikely that two independent genomic k-mers will have the same sequence in all but one nucleotide simply by chance. Thus, the two k-mers in a k-mer pair are homologous and can either represent different alleles of the same locus (heterozygous k-mers) or different loci (paralogs, e.g., duplicated genes or transposable elements). In a reasonably heterozygous genome, the signal from heterozygous k-mers will dominate and therefore can be used to generate an estimate of ploidy.

We denote the two k-mers in each k-mer pair as $A$ and $B$ such that the coverage of $A$ (CovA) is always greater than or equal to the coverage of $B$ (CovB). Within every pair, both $A$ and $B$ can be present in one or more genomic copies and

therefore CovA + CovB $\in \{2\lambda, 3\lambda, 4\lambda, 5\lambda, ...\}$, where $\lambda$ is the monoploid genome coverage. Plotting CovA + CovB versus $\frac{CovB}{CovA+CovB}$ will result in each distinct genomic structure projecting on a different position (i.e., "smudge") in 2D space (see Fig. 7).

By plotting the total coverage of the k-mer pair, CovA + CovB, versus the relative minor k-mer coverage, $\frac{CovB}{CovA+CovB}$, we can identify individual "smudges" that correspond to different haplotype structures. Due to the Poisson nature of the coverages of each position along the genome that is typical in sequencing experiments, the k-mer pairs will not have the exact coordinates as given in Fig. 7. However, it is usually possible to resolve the smudge to which each pair belongs. Figure 8a shows an ideal case, where the sequencing coverage is sufficient to completely separate all the smudges, providing very strong evidence of triploidy. The brightness of each smudge is determined by the number of k-mer pairs that fall within it.

The Smudgeplot estimates of monoploid coverage and ploidy allow users to visualize and discover properties about genomes with high levels of imperfect duplications, various ploidy levels, and high heterozygosity (see Supplementary Methods for details). Smudgeplot provides users with a results table that indicates the number of k-mer pairs that fall within each annotated smudge. We recommend using these values in addition to the ploidy estimate to help determine the structure of the genome. Smudgeplot is a visualization tool that is especially powerful in combination with GenomeScope, as both independently estimate monoploid coverage by exploiting different genomic properties. Notably, Smudgeplot is able to accurately predict that *Fragaria x ananassa* is octaploid (see Fig. 8b).

## Data availability
Genuine sequencing data are available using the accession codes listed in Supplementary Table 1. The code and parameters used for generating the simulated datasets are available in the GenomeScope 2.0 GitHub repository. The full results of modeling the simulated datasets are available in Supplementary Data 1.

## Code availability
All code supporting the current study is deposited in GitHub at https://github.com/tbenavi1/genomescope2.0 and https://github.com/KamilSJaron/smudgeplot. Permanent repositories are available at https://doi.org/10.5281/zenodo.3657798 and https://doi.org/10.5281/zenodo.3658220. We also have a web-enabled version of GenomeScope available at http://genomescope.org/genomescope2.0/.

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

## Acknowledgements
We would like to thank Edward Scheinerman for his thorough explanation of combinatorics topics. We would also like to thank Tanja Schwander and Marc Robinson-Rechavi for their helpful discussions. This work was supported, in part, by NIH grant R01-HG006677 and NSF grants DBI-1350041 and IOS-1732253 to M.C.S. K.S.J. was supported by Swiss National Foundation grant CRSII3_160723. Part of this research project was conducted using computational resources at the Maryland Advanced Research Computing Center (MARCC).

## Author contributions
T.R.R.-B. extended the GenomeScope model for polyploid genomes. T.R.R.-B. and K.S.J. conceived and implemented Smudgeplot. M.C.S. supervised the project. All authors wrote the manuscript.

## Competing interests
The authors declare no competing interests.

**Additional information**

