## [Peer Review File · Nature Communications]

Reviewers' Comments:

Reviewer #1:

Remarks to the Author:

Dear Colleagues,

I read your manuscript describing methods to evaluate genome polyploidy, repetitiveness and size ahead of de novo assembly with great interest. You present two tools, GenomeScope 2.0, an improvement on v1.0 which handles polyploid genomes, and SmudgePlots, a visualization tool to help determine said polyploidy. As you explain in the introduction, these parameters are paramount to genome assembly and quality control, hence developing these tools will be important as we sequence more and more species. The results your present indicate a robust method.

The text was however somewhat unclear with regards to the methodology. In many cases, definitions were provided as examples, which raised questions in my mind as to how they generalised:

- Regarding SmudgePlots the description at the bottom of page 6 and top of left me unsatisfied, in particular with the labelling of the smudges. E.g. you mention the smudge with minor relative coverage of 0.5 and lowest sum of coverages is labelled AB, but what if that smudge has a relative coverage of 0.47? How do you label all other smudges? Could you summarise the algorithm into pseudocode?
- Regarding GenomeScope, why is foreknowledge of ploidy necessary? Could you not run the model with a high p (e.g. 10) then see whether some of the ploidy levels have very weak alphas?
- In the supplement, could you please provide more information on the calculation of $r_{\{\phi\}}$ in general? E.g. how does one calculate $r_{\{(3,2,1)\}}$?
- Page S5: How does the infinite sites model come into play? You give the example of triallelic a position (aaabbc), surely that violates the assumption that two independent mutations can't hit the same site?
- Page S7, equation (12): α_i has a fairly hefty equation to evaluate: it contains a sum with a combinatorial number of terms that each rely on the evaluation of s_{ϕ} , which itself is a sum over many terms that contain r . How exactly do you use this equation to determine the parameters? Does nlsLM simply set the parameters on these complex expressions?

Miscellaneous points:

- Page 2, bottom: "The relative heights of the peaks are proportional to the heterozygosity of the species" This is unclear to me. For example, on Figure 1(b) there are 3 peaks, how do I infer heterozygosity from that?
- Page 2, bottom: Figure 1(b) is meant to be an example triploid genome, but the 3rd peak is tiny with respect to the other two. How do we know the genome is triploid from this plot?
- Page 3: "the i -th peak corresponds to the contributions from k -mers that appear approximately i times in the polyploid genome". Firstly this is confusing because this is written under the diploid model section. Secondly, surely the number of occurrences in the genome is not approximate? I.e. the first peak corresponds to k -mers that appear exactly once in the genome.
- Page 4: top line: "are derived () is called" (the comma is standing alone between subject and verb).

- Page 5-7: You switched the minority allele from A (on page 5) to B (Figure 3). Figure 2 is split because the column legend has A as the minor allele, but the labels in column 1 have B as the major allele.

- Page 7: I have never met the notation AAAA -> AAAB -> AABC -> ABCD to describe a tree topology (I'm personally much more used to the Newick notation, but that could just be me). Could you please provide a reference explaining how to interpret this notation?

- Page S5: "the k-mer nucleotide heterozygosity rate" I'm guessing the word "nucleotide" is too many here?

Reviewer #2:

Remarks to the Author:

Ranallo-Benavidez et al. present GenomeScope 2.0, an extension of the popular GenomeScope method to polyploid genomes. The underlying mathematical model is a conceptually straightforward generalization of the diploid case; however, solving the implied equation systems requires some relatively involved operations, such as the Möbius inversion formula. In addition to GenomeScope 2.0, the authors present Smudgeplots, an approach to obtain a ploidy estimate in a reference-free manner by examination of heterozygous kmer frequencies. Of note, application of GenomeScope 2.0 requires a ploidy estimate, so that Smudgeplots and GenomeScope 2.0 can work together nicely.

The paper is well written and very clear. The structure of the paper is easy to follow. Validation of the methods is done, first, by carrying out a comprehensive range of simulation experiments; second, by application to real datasets. The chosen validation datasets are interesting and represent a variety of organisms. Taken together, simulated and real-data experiments show that GenomeScope 2.0 is a highly accurate method for the reference-free analysis of polyploid genomes. As argued convincingly by the authors in the Introduction, having such a method can be useful for a variety of purposes, ranging from an initial characterization of a genome from unassembled sequencing data to providing orthogonal QC metrics for assembly. I think that GenomeScope 2.0 fulfills an important community need and strongly support publication.

I only have a few comments on this very strong paper:

1) Section 3.2 on the transformed k-mer histogram: It reads as if the transformed histograms were analysed exactly like the non-transformed one, i.e. as if the equations of the mathematical model were not adjusted for the fact that e.g. log-ed data is being analysed. Is this correct?

2) It seems that the simulations don't cover the case of ploidy = 2. It seems relatively obvious that GenomeScope 2.0 will work well for the standard diploid case, but for the sake of completeness and for the sake of having a single tool validated on a complete range of possible ploidies, adding the standard diploid case might be good.

3) Is the ploidy estimator of Smudgeplots also validated in the simulation experiments? I might have missed the corresponding passages – but if not, why not? I assume that the validation of GenomeScope assumes that true ploidy is known - in addition, it would be good to measure performance when ploidy is also estimated from the simulated data.

4) Would any summary statistics indicate poor model fit? In the original GenomeScope paper, the authors discuss examining the residuals of the model and the number of unexplained kmers - I suppose something along these lines would also be possible for GenomeScope 2.0? The authors could consider adding a short section on indicators of model failure to the Discussion section.

Reviewer #3:

Remarks to the Author:

The authors report on a new, enhanced version of GenomsScope, a tool for examining kmer plots of genomes which also fits a diploid/haploid model. They also detail a new tool called Smudgeplot that plots kmer pairs to give a 2d, visual representation of pairs that differ by 1 base in the kmer and estimates single haplotype coverage.

The manuscript is presented well and the tools are likely very useful to look at unknown genomes (kmer plots have been used for many years, but genomescope adds an ease of analysis). The examples presented here are informative for the tools and I appreciate the simulation data.

This software provides a nice way of viewing this type of data with potential applications for people who work on unknown or complex genomes.

There are a few more critical issues:

1. These are specialized tools, likely to be used by a very small number of people. It is hard to suggest that this will be of wide use to readers of Nature Communication. It is very possible that Smudgeplots could be useful for specialized applications. A good way to think about this is – there are probably 1000 people in the world who have generated a kmer plot from an Illumina sample. Of those probably 100 have generated more than 1. Of those the number who work in complex plants (or complex vertebrate) that might benefit from being able to visualize the het rate and ploidy are probably around 10.

2. This manuscript is fundamentally written as a CS manuscript (similar to the first GenomeScope paper published in Bioinformatics). This detracts from the impact in a more general journal. It is very inaccessible outside of a limited group. There are great places to send this kind of manuscript and work that is not NC.

3. If the manuscript were reformatted to provide a more general view of the tools and the usefulness, there would still be the problem of non-experts in this area being able to run and interpret the results. The software has many dependencies that are not explicit in the installation instructions and running the software to process kmers takes a serious amount of RAM with the current python implementation to run. It also has serious run time associated with it. These could be improved with streamlined implementation and refactoring to make this more generally useable. My group is able to follow the large collection of varied examples leading to varied outcomes and varied interpretations from the smudgeplots. But even after spending time puzzling over the interpretation, I wouldn't feel confident in extrapolating an evolutionally scenario and consequences on ploidy and heterozygosity from a smudgeplot. Its feels too variable to me- for example take the case of a hexploid genome, where one subgenome has little to no heterozygosity and 1 subgenome has "normal" levels that split on the smudgeplot and the final subgenome is highly divergent that breaks up the kmer pairs. I feel like we'd look at the plot and say, something seems messed up with this genome.

How do Smudge plots perform with CCS/"Hi-Fi" data? This data type has built in 1 off kmer pairs because of the systematic errors.

In general, I think this could be a useful tool for characterizing complex genomes, but it has limited users and could benefit from software improvements and a way to reduce the learning curve for interpretation. As it stands now, you need to be an expert to manage to get it to run, run it properly, and to interpret the results.

Reviewers' comments:

Reviewer #1 (Remarks to the Author):

Dear Colleagues,

I read your manuscript describing methods to evaluate genome polyploidy, repetitiveness and size ahead of de novo assembly with great interest. You present two tools, GenomeScope 2.0, an improvement on v1.0 which handles polyploid genomes, and SmudgePlots, a visualization tool to help determine said polyploidy. As you explain in the introduction, these parameters are paramount to genome assembly and quality control, hence developing these tools will be important as we sequence more and more species. The results your present indicate a robust method.

The text was however somewhat unclear with regards to the methodology. In many cases, definitions were provided as examples, which raised questions in my mind as to how they generalised:

- Regarding SmudgePlots the description at the bottom of page 6 and top of left me unsatisfied, in particular with the labelling of the smudges. E.g. you mention the smudge with minor relative coverage of 0.5 and lowest sum of coverages is labelled AB, but what if that smudge has a relative coverage of 0.47? How do you label all other smudges? Could you summarise the algorithm into pseudocode?

We have added a supplementary section to better explain the Smudgeplot method along with pseudocode in a concise format. We also included new benchmarking results showing the accuracy for Smudgeplot over a wide range of genome structures. Briefly, the method considers “smudges” whose minor coverage ratios fall within 0.01 of expected values ($\frac{1}{2}$, $\frac{1}{3}$, etc.). Consequently, a “smudge” with a relative coverage of 0.47 would not get annotated with our

method. However, as the peaks are automatically determined by the data this could only occur if the x coordinate of the first peak of the kmer spectra is 0.47 times that of the second peak. This is a highly unusual composition (e.g. aneuploid sample, insufficient data quality with very high error or low coverage sequencing data) that are not supported by the method. Importantly, we do not observe such a scenario in any of our genuine or simulated datasets.

- Regarding GenomeScope, why is foreknowledge of ploidy necessary? Could you not run the model with a high p (e.g. 10) then see whether some of the ploidy levels have very weak alphas?

If GenomeScope is run with an incorrect value of p that is greater than the true value it will fit a greater number of negative binomial distributions than is necessary. This can result in GenomeScope “overfitting”, and converging to incorrect values. We have added the following text to the manuscript in the discussion section:

“It is important to run GenomeScope with the correct value for the ploidy parameter. If p is greater than the true value this can lead to overfitting where the model contains a greater number of negative binomial distributions than is necessary. In real polyploid data, especially for highly heterozygous and repetitive genomes, it can be ambiguous whether smaller higher-order peaks truly represent repeats or higher ploidy.”

- In the supplement, could you please provide more information on the calculation of $r_{\{\phi\}}$ in general? E.g. how does one calculate $r_{\{3,2,1\}}$?

The values of $r_{\{\phi\}}$ are parameters that are estimated by GenomeScope 2.0 directly by the non-linear optimization algorithm relative to the mixture model representing the given ploidy level. They are not “calculated” per se (unlike derived values α_i or s_i). We have added the following text to the manuscript:

“These nucleotide homozygosity rates are the parameters that are estimated by GenomeScope 2.0 through the non-linear optimization algorithm.”

- Page S5: How does the infinite sites model come into play? You give the example of triallelic a position (aaabbc), surely that violates the assumption that two independent mutations can't hit the same site?

We have edited the manuscript to be more precise about the assumptions inherent in the model. Specifically, we added the following text to the manuscript:

“For our model we make the following assumptions: 1) each locus of the genome is independent of the other loci and 2) the nucleotide heterozygosity rates are constant over the entire genome. Unlike the infinite sites model, our model does not assume that every new mutation must occur at a new site especially since the model analyzes kmers rather than individual nucleotides.”

- Page S7, equation (12): α_i has a fairly hefty equation to evaluate: it contains a sum with a combinatorial number of terms that each rely on the evaluation of s_φ , which itself is a sum over many terms that contain r . How exactly do you use this equation to determine the parameters? Does nlsLM simply set the parameters on these complex expressions?

In the `model_functions.R` file, we have explicitly written out the combinatorial number of terms for each equation. Then, nlsLM is used to determine the parameters that minimize the residual sum of squares between the model and the real data.

We have added the following text to the manuscript:

“For each ploidy up to $p=6$, we have explicitly written in the code the many terms for the equations for α_i and s_{φ} . Then, non-linear optimization is used to determine the parameters that minimize the residual sum of squares between the model and the real data. GenomeScope 2.0 currently only supports analyzing organisms with ploidy up to 6, due to the combinatorial number of terms in these equations.”

Miscellaneous points:

- Page 2, bottom: "The relative heights of the peaks are proportional to the heterozygosity of the species" This is unclear to me. For example, on Figure 1(b) there are 3 peaks, how do I infer heterozygosity from that?

Thank you for pointing this out -- Figure 1 demonstrates there are additional peaks in higher ploidy samples, but you have correctly pointed out that this does not demonstrate the role of the heights of the peaks. We have revised this section to clarify this point. The plots in Supplemental Section S5 show how higher rates of heterozygosity impact the peak heights:

“For example, for a diploid species, increasing heterozygosity will result in a higher first peak and a lower second peak. For a polyploid species, the relationship is more complicated, but in general increasing heterozygosity will result in a higher first peak and lower subsequent peaks.”

- Page 2, bottom: Figure 1(b) is meant to be an example triploid genome, but the 3rd peak is tiny with respect to the other two. How do we know the genome is triploid from this plot?

This figure was meant to serve as a demonstration of how the kmer profile can look for a known triploid species with three recognizable peaks rather than two in the diploid example. We added the following text to the manuscript:

“Occasionally, it is difficult to determine whether a peak in the k-mer spectrum is a major peak. For this reason, GenomeScope 2.0 analyzes a transformed k-mer spectrum (see Section 3.2) in

which the heights of higher-order peaks are increased. If the ploidy is still uncertain the user may run our Smudgeplot tool (see Section 4).”

- Page 3: "the i -th peak corresponds to the contributions from k -mers that appear approximately i times in the polyploid genome". Firstly this is confusing because this is written under the diploid model section. Secondly, surely the number of occurrences in the genome is not approximate? I.e. the first peak corresponds to k -mers that appear exactly once in the genome.

We edited the text to say “exactly” instead of “approximately” and “diploid” instead of “polyploid” since this is written under the diploid model section.

- Page 4: top line: "are derived () is called" (the comma is standing alone between subject and verb).

We have removed the unnecessary comma.

- Page 5-7: You switched the minority allele from A (on page 5) to B (Figure 3). Figure 2 is split because the column legend has A as the minor allele, but the labels in column 1 have B as the major allele.

We have edited the manuscript to consistently use A as the major allele and B as the minor allele. In particular, we have edited the text of the manuscript, as well as figure 2 labels and legends to be consistent.

- Page 7: I have never met the notation AAAA -> AAAB -> AABC -> ABCD to describe a tree topology (I'm personally much more used to the Newick notation, but that could just be me). Could you please provide a reference explaining how to interpret this notation?

We have edited the manuscript to utilize the Newick notation, since this is a more standard way to express phylogenetic trees.

- Page S5: "the k -mer nucleotide heterozygosity rate" I'm guessing the word "nucleotide" is too many here?

We have removed the word “nucleotide.”

Best regards,

Daniel Zerbino

Reviewer #2 (Remarks to the Author):

Ranallo-Benavidez et al. present GenomeScope 2.0, an extension of the popular GenomeScope method to polyploid genomes. The underlying mathematical model is a conceptually straightforward generalization of the diploid case; however, solving the implied equation systems requires some relatively involved operations, such as the Möbius inversion formula. In addition to GenomeScope 2.0, the authors present Smudgeplots, an approach to obtain a ploidy estimate in a reference-free manner by examination of heterozygous kmer frequencies. Of note, application of GenomeScope 2.0 requires a ploidy estimate, so that Smudgeplots and GenomeScope 2.0 can work together nicely.

The paper is well written and very clear. The structure of the paper is easy to follow. Validation of the methods is done, first, by carrying out a comprehensive range of simulation experiments; second, by application to real datasets. The chosen validation datasets are interesting and represent a variety of organisms. Taken together, simulated and real-data experiments show that GenomeScope 2.0 is a highly accurate method for the reference-free analysis of polyploid genomes. As argued convincingly by the authors in the Introduction, having such a method can be useful for a variety of purposes, ranging from an initial characterization of a genome from unassembled sequencing data to providing orthogonal QC metrics for assembly. I think that GenomeScope 2.0 fulfills an important community need and strongly support publication.

Thank you for your comments.

I only have a few comments on this very strong paper:

1) Section 3.2 on the transformed k-mer histogram: It reads as if the transformed histograms were analysed exactly like the non-transformed one, i.e. as if the equations of the mathematical model were not adjusted for the fact that e.g. log-ed data is being analysed. Is this correct?

The equation of the model is adjusted accordingly. To be more explicit, we have added the following text to the manuscript:

“... and the mathematical equation for the model used during the non-linear optimization is adjusted accordingly.”

2) It seems that the simulations don't cover the case of ploidy = 2. It seems relatively obvious that GenomeScope 2.0 will work well for the standard diploid case, but for the sake of completeness and for the sake of having a single tool validated on a complete range of possible ploidies, adding the standard diploid case might be good.

For the sake of completeness we have added simulations for both ploidy = 2 and ploidy = 1. The text and figures of the paper have been updated accordingly. We also added the following

paragraph comparing the accuracy and robustness of GenomeScope 2.0 versus GenomeScope 1.0:

“When compared to GenomeScope 1.0, GenomeScope 2.0 is more robust and accurate, especially on lower coverage diploid data. Specifically, GenomeScope 1.0 failed to converge for 35 of the 51 simulated heterozygosity datasets, converged to the wrong peak due to low sequencing coverage for 15 of the datasets, and produced accurate results for only 1 dataset. GenomeScope 1.0 failed to converge on 2 of the 51 simulated repetitiveness datasets and converged to the wrong peak for the other 49 datasets. Lastly, GenomeScope 1.0 failed to converge for 3 of the 4 simulated length datasets and produced inaccurate results for the other dataset. Based on these results, we encourage all users to use GenomeScope 2.0 for diploid datasets.”

3) Is the ploidy estimator of Smudgeplots also validated in the simulation experiments? I might have missed the corresponding passages – but if not, why not? I assume that the validation of GenomeScope assumes that true ploidy is known - in addition, it would be good to measure performance when ploidy is also estimated from the simulated data.

We have expanded section 5.1 on the simulated results to include results for Smudgeplot. We find that Smudgeplot accurately estimates ploidy over a broad range of genomic compositions (ploidy, heterozygosity, repetitiveness) although can fail for extreme values of heterozygosity or repetitiveness. In these more extreme scenarios the composition of the peaks become ambiguous so that some kmer pairs cannot be identified and quantified. We acknowledge the range of confident assessments in the discussion and added the details to section 5.1:

“Finally, we validated Smudgeplot on simulated data. In each case, we simulated 25x coverage per homologue and 1% sequencing error using a random 10 Mbp monoploid genome as a “progenitor.” We simulated both the allotetraploid and autotetraploid topologies for ploidy 4, and a single topology for ploidies 2, 3, 5, and 6. For nucleotide divergence, we systematically evaluated across 0.5% to 25% in 0.5% increments while holding the repetitiveness constant at 10%, for a total of 50 values. For repetitiveness, we evaluated a parameter sweep from 0% to 50% in 1% increments while holding the nucleotide divergence constant at 2.0%, for a total of 51 values. For the full results, see Section S3.”

“For the nucleotide divergence sweep, Smudgeplot correctly estimates ploidy for the diploid simulated data over all heterozygosity values, for the triploid data up to 24.0% heterozygosity, for the allotetraploid data up to 18.0% heterozygosity, for the autotetraploid data up to 23.5% heterozygosity, for the pentaploid data up to 24.0% heterozygosity, and for the hexaploid data up to 24.0% heterozygosity. Above these heterozygosity thresholds, Smudgeplot underestimates the ploidy due to the k-mers in a k-mer pair being more than one nucleotide different and thus not identified.”

“For the repetitiveness sweep, Smudgeplot correctly estimates ploidy for the diploid simulated data up to 39% repetitiveness, for the triploid data up to 38% repetitiveness, for the allotetraploid data up to 43% repetitiveness, for the autotetraploid data up to 38% repetitiveness, for the pentaploid data over all repetitiveness values, and for the hexaploid data over all repetitiveness values. Above these repetitiveness thresholds, Smudgeplot overestimates the ploidy to the signal from repetitive k-mers dominating the signal from heterozygous k-mers.”

4) Would any summary statistics indicate poor model fit? In the original GenomeScope paper, the authors discuss examining the residuals of the model and the number of unexplained kmers - I suppose something along these lines would also be possible for GenomeScope 2.0? The authors could consider adding a short section on indicators of model failure to the Discussion section.

In addition to describing the failure mode when running with an incorrect ploidy parameter (as described above), we have added the following text to the Discussion section to comment on other common issues:

“In order to accurately estimate genome size for highly repetitive genomes, it is important to create a k-mer histogram that is not truncated. By default, KMC and Jellyfish truncate the histogram at 10,000. We suggest running these tools without a maximum counter. The model fit that is output by GenomeScope can also be used to identify poor fit or incomplete datasets. However, in general, the best indicator of a good model fit is inspecting the plots to ensure the model matches the empirical data across the distribution.”

Reviewer #3 (Remarks to the Author):

The authors report on a new, enhanced version of GenomsScope, a tool for examining kmer plots of genomes which also fits a diploid/haploid model. They also detail a new tool called Smudgeplot that plots kmer pairs to give a 2d, visual representation of pairs that differ by 1 base in the kmer and estimates single haplotype coverage.

The manuscript is presented well and the tools are likely very useful to look at unknown genomes (kmer plots have been used for many years, but genomescope adds an ease of analysis). The examples presented here are informative for the tools and I appreciate the simulation data.

This software provides a nice way of viewing this type of data with potential applications for people who work on unknown or complex genomes.

Thank you for your comments.

There are a few more critical issues:

1. These are specialized tools, likely to be used by a very small number of people. It is hard to suggest that this will be of wide use to readers of Nature Communication. It is very possible that Smudgeplots could be useful for specialized applications. A good way to think about this is – there are probably 1000 people in the world who have generated a kmer plot from an Illumina sample. Of those probably 100 have generated more than 1. Of those the number who work in complex plants (or complex vertebrate) that might benefit from being able to visualize the het rate and ploidy are probably around 10.

We respectfully disagree with your assessment. In 2017, we published GenomeScope exclusively for diploid genomes, and since then it has been cited 181 times (Google Scholar on Dec 19) and has accumulated over 25,000 unique runs at genomescope.org. Additionally, a major feature request from several users has been to support higher ploidy genomes, especially for resequencing major crop species such as wheat and potatoes. GenomeScope has also become a core component to major genomics initiatives, such as the Vertebrate Genome Project, that routinely use this method for hundreds of samples a year. Finally, thanks to the improved model fitting algorithm and other enhancements, this improved version of GenomeScope will become the new default version for all ploidy values. So while we cannot guarantee that this improved version will attract such attention, we anticipate considerable use from users studying diverse organisms across the tree of life.

2. This manuscript is fundamentally written as a CS manuscript (similar to the first GenomeScope paper published in Bioinformatics). This detracts from the impact in a more general journal. It is very inaccessible outside of a limited group. There are great places to send this kind of manuscript and work that is not NC.

We believe this is the most appropriate format and venue for the method. The methods are sophisticated, yet in the main text we have presented a high level overview of the methods so that potential users will be exposed to the general concepts and then quickly move on to results validating the method with real and simulated data. To further clarify the presentation, we moved several paragraphs of the Smudgeplot methods to the supplement.

3. If the manuscript were reformatted to provide a more general view of the tools and the usefulness, there would still be the problem of non-experts in this area being able to run and interpret the results. The software has many dependencies that are not explicit in the installation instructions and running the software to process kmers takes a serious amount of RAM with the current python implementation to run. It also has serious run time associated with it. These could be improved with streamlined implementation and refactoring to make this more generally useable.

GenomeScope 2.0 can be run via a simple web interface after running a few simple command line tools. We have improved Smudgeplot by documenting the dependencies and streamlining the operations. Notably, now users can install Smudgeplot using the command “conda install -c bioconda smudgeplot”. Additionally, we have implemented a C++ version of Smudgeplot which utilizes the KMC API to substantially reduce the computational requirements. For this, we have forked the repo of KMC and created a new smudge_pairs program which allows users to find k-mer pairs directly from the binary compressed database. With this implementation, Smudgeplot runs on the order of hours instead of on the order of days for several of the genomes we tested. Please see <https://github.com/KamilSJaron/smudgeplot/tree/dev2> for this new version of Smudgeplot. We will release this as the new version once the review of this manuscript has been completed. For the python implementation, users will need a fairly sizeable amount of RAM for some steps of the analysis. However, considering that most users will run these tools in support of de novo genome assembly, these users are likely to have access to large memory servers.

My group is able to follow the large collection of varied examples leading to varied outcomes and varied interpretations from the smudgeplots. But even after spending time puzzling over the interpretation, I wouldn't feel confident in extrapolating an evolutionary scenario and consequences on ploidy and heterozygosity from a smudgeplot.

It feels too variable to me- for example take the case of a hexploid genome, where one subgenome has little to no heterozygosity and 1 subgenome has “normal” levels that split on the smudgeplot and the final subgenome is highly divergent that breaks up the kmer pairs. I feel like we'd look at the plot and say, something seems messed up with this genome.

With the validated results on genuine sequencing data along with the added section on Smudgeplot simulated results, we document that Smudgeplot is effective over a broad range of biologically interesting parameters. However, we also acknowledge that more extreme genome structures can remain ambiguous and encourage users to look at the number of k-mer pairs that fall within each smudge as well as at the ploidy estimate to evaluate the confidence in the prediction. For your example, there indeed may be ambiguity since heterozygosity near zero does cause kmers to fall entirely within the homozygous peak, and very high heterozygosity (with multiple substitutions per kmer) causes those the kmers to fall within the homozygous peak as we discuss in the manuscript.

However, we highlight that such ambiguity is present for essentially all genomics analysis: de novo assembly, read mapping, variant calling, differential expression, peak calling, etc all fail if the data characteristics are outside of an expected range (too low or too high of coverage, extreme levels of repeats/multimapping reads, extreme levels of sequencing errors, extreme levels of heterozygosity, etc). Nevertheless, these tools are still useful for many researchers when the range of conditions is documented and explained as we have done here.

We added the following text to the manuscript when discussing the interpretation of the results:

“Smudgeplot provides users with a results table that indicates the number of k-mer pairs that fall within each annotated smudge. We recommend using these values in addition to the ploidy estimate to help determine the structure of the genome.”

How do Smudge plots perform with CCS/“Hi-Fi” data? This data type has built in 1 off kmer pairs because of the systematic errors.

As stated in the manuscript, we have designed and implemented these tools for low error Illumina sequencing. Future work remains to validate it as public polyploid HiFi data becomes available. We agree that any systematic errors could potentially skew the results.

In general, I think this could be a useful tool for characterizing complex genomes, but it has limited users and could benefit from software improvements and a way to reduce the learning curve for interpretation. As it stands now, you need to be an expert to manage to get it to run, run it properly, and to interpret the results.

We also believe this will be a useful tool, as have our early test users. We have improved the description and interface of both methods according to the feedback of you and our users. Although we agree that the interpretation is not always straightforward, the installation and usability has improved a lot since the first release. We have also added to the discussion to highlight potential failure modes for the algorithm, especially extreme values of heterozygosity or repetitiveness that introduce ambiguity to the results.

Reviewers' Comments:

Reviewer #1:

Remarks to the Author:

Dear Colleagues,

Thank you for responding to my questions and comments.

Reviewer #2:

Remarks to the Author:

Ranallo-Benavidez et al. have now submitted a revised version of the GenomeScope 2.0 paper.

The revisions include i) an improved description of the Smudgeplots method; ii) more comprehensive evaluations based on simulated data, in particular covering the ploidy = 1/2 cases and the Smudgeplots method - interestingly showing that GenomeScope 2.0 outperforms the original GenomeScope on e.g. the diploid base case.

I am happy with the revisions and would support the publication of the papers in its present form!
Nice job!

Reviewer #3:

Remarks to the Author:

1. Specialized tools discussion

"We respectfully disagree with your assessment. In 2017, we published GenomeScope exclusively for diploid genomes, and since then it has been cited 181 times (Google Scholar on Dec 19) and has accumulated over 25,000 unique runs at genomescope.org . Additionally, a major feature request from several users has been to support higher ploidy genomes, especially for resequencing major crop species such as wheat and potatoes. GenomeScope has also become a core component to major genomics initiatives, such as the Vertebrate Genome Project, that routinely use this method for hundreds of samples a year. Finally, thanks to the improved model fitting algorithm and other enhancements, this improved version of GenomeScope will become the new default version for all ploidy values. So while we cannot guarantee that this improved version will attract such attention, we anticipate considerable use from users studying diverse organisms across the tree of life. "

You might want to look at the distribution of users who have used the web version of GenomeScope. The unique users are likely to be much smaller than you believe from total gross numbers of histogram runs. Regardless 25k histograms and 111 citations defiantly shows that this is useful tool (Jellyfish which is arguably a more general purpose tool and needed to run GenomeScope has 746 citations (with ~500 since 2017) . But GenomeScope is still a specialty tool that is used by a small number of people. I do not believe that an update to the original Bioinformatics paper is suitable for the wider audience of Nature Communication and it may not have much useful impact over the original paper. It seems that you are taking a relatively obscure tool and then saying it will be applicable to a broader audience because you have made it even more specialized! Maybe super useful for a small number of people or more useful in the future. Maybe.

2. SmudegPlot difficult to install and run – extremely limiting for larger genomes

"Notably, now users can install Smudgeplot using the command "conda install -c bioconda smudgeplot". Additionally, we have implemented a C++ version of Smudgeplot which utilizes the KMC API to substantially reduce the computational requirements. For this, we have forked the repo of KMC and created a new smudge_pairs program which allows users to find k-mer pairs directly from the binary compressed database. With this implementation, Smudgeplot runs on the order of hours instead of on the order of days for several of the genomes we tested. Please see <https://github.com/KamilSJaron/smudgeplot/tree/dev2> for this new version of Smudgeplot."

This is great advance over the previous work- I believe these changes will make it tool that can be used.

3. To follow up on discussion of applicability (including CCS/"Hi-Fi")

Will these tools be used in the future outside of large screening efforts (like as noted in response the V10k) or graduate students who are looking at older data? Most genome production has already moved to PACBIO/ONT based systems, and with costs coming down (at least for PACBIO data) will people even generate Illumina data up front?

REVIEWERS' COMMENTS:

Reviewer #1 (Remarks to the Author):

Dear Colleagues,

Thank you for responding to my questions and comments.

Best regards,

Daniel Zerbino

Thanks for your comments.

Reviewer #2 (Remarks to the Author):

Ranallo-Benavidez et al. have now submitted a revised version of the GenomeScope 2.0 paper.

The revisions include i) an improved description of the Smudgeplots method; ii) more comprehensive evaluations based on simulated data, in particular covering the ploidy = 1/2 cases and the Smudgeplots method - interestingly showing that GenomeScope 2.0 outperforms the original GenomeScope on e.g. the diploid base case.

I am happy with the revisions and would support the publication of the papers in its present form! Nice job!

Thanks for your comments.

Reviewer #3 (Remarks to the Author):

1. Specialized tools discussion

“We respectfully disagree with your assessment. In 2017, we published GenomeScope exclusively for diploid genomes, and since then it has been cited 181 times (Google Scholar on Dec 19) and has accumulated over 25,000 unique runs at genomescope.org . Additionally, a major feature request from several users has been to support higher ploidy genomes, especially for resequencing major crop species such as wheat and potatoes. GenomeScope has also become a core component to major genomics initiatives, such as the Vertebrate Genome Project, that routinely use this method for hundreds of samples a year. Finally, thanks to the improved model fitting algorithm and other enhancements, this improved version of GenomeScope will become the new default version for all ploidy values. So while we cannot guarantee that this improved version will attract such attention, we anticipate considerable use from users studying diverse organisms across the tree of life. “

You might want to look at the distribution of users who have used the web version of GenomeScope. The unique users are likely to be much smaller than you believe from total gross numbers of histogram runs. Regardless 25k histograms and 111 citations defiantly shows that this is useful tool (Jellyfish which is arguably a more general purpose tool and needed to run GenomeScope has 746 citations (with

~500 since 2017) . But GenomeScope is still a specialty tool that is used by a small number of people. I do not believe that an update to the original Bioinformatics paper is suitable for the wider audience of Nature Communication and it may not have much useful impact over the original paper. It seems that you are taking a relatively obscure tool and then saying it will be applicable to a broader audience because you have made it even more specialized! Maybe super useful for a small number of people or more useful in the future. Maybe.

2. SmudgePlot difficult to install and run – extremely limiting for larger genomes

“Notably, now users can install Smudgeplot using the command “conda install -c bioconda smudgeplot”. Additionally, we have implemented a C++ version of Smudgeplot which utilizes the KMC API to substantially reduce the computational requirements. For this, we have forked the repo of KMC and created a new smudge_pairs program which allows users to find k-mer pairs directly from the binary compressed database. With this implementation, Smudgeplot runs on the order of hours instead of on the order of days for several of the genomes we tested. Please see <https://github.com/KamilSJaron/smudgeplot/tree/dev2> for this new version of Smudgeplot.”

This is great advance over the previous work- I believe these changes will make it tool that can be used.

3. To follow up on discussion of applicability (including CCS/“Hi-Fi”)

Will these tools be used in the future outside of large screening efforts (like as noted in response the V10k) or graduate students who are looking at older data? Most genome production has already moved to PACBIO/ONT based systems, and with costs coming down (at least for PACBIO data) will people even generate Illumina data up front?

Thank you for your comments. We have expanded the discussion to explicitly explain that we currently only support low error reads. We also believe that there remains a huge need for Illumina data for polishing in combination with long reads.